# Completion of Genetic Testing and Incidence of Pathogenic Germline Mutation among Patients with Early-Onset Colorectal Cancer: A Single Institution Analysis

**DOI:** 10.3390/cancers15143570

**Published:** 2023-07-11

**Authors:** Michael H. Storandt, Kara R. Rogen, Anushka Iyyangar, Rylie R. Schnell, Jessica L. Mitchell, Joleen M. Hubbard, Frank A. Sinicrope, Aminah Jatoi, Amit Mahipal, Qian Shi, Zhaohui Jin

**Affiliations:** 1Department of Medicine, Mayo Clinic, Rochester, MN 55905, USA; storandt.michael@mayo.edu; 2Department of Clinical Genomics, Mayo Clinic, Rochester, MN 55905, USA; rogen.kara@mayo.edu; 3Department of Oncology, Mayo Clinic, Rochester, MN 55905, USA; iyyangar.anushka@mayo.edu (A.I.); schnell.rylie@mayo.edu (R.R.S.); mitchell.jessica@mayo.edu (J.L.M.); hubbard.joleen@mayo.edu (J.M.H.); sinicrope.frank@mayo.edu (F.A.S.); jatoi.aminah@mayo.edu (A.J.); 4Department of Oncology, University Hospitals Seidman Cancer Center and Case Western Reserve University, Cleveland, OH 44106, USA; amit.mahipal@uhhospitals.org; 5Department of Quantitative Health Sciences, Mayo Clinic, Rochester, MN 55905, USA; shi.qian2@mayo.edu

**Keywords:** early-onset, colorectal cancer, germline testing, pathogenic variant

## Abstract

**Simple Summary:**

Rates of early-onset colorectal cancer (eoCRC), defined as <50 years of age at diagnosis, have increased. Nearly one-quarter of cases of eoCRC may be associated with a germline pathogenic variant (PGV) resulting in a hereditary cancer syndrome. In the present study, we reviewed patients with a history of colorectal cancer who were referred to medical genetics at our institution to better understand the prevalence and spectrum of PGVs in both patients with eoCRC as well as with average-onset CRC (aoCRC). We found that approximately one in four patients with eoCRC had a PGV, which included 8.3% with Lynch syndrome. In patients with aoCRC, similar rates of detection of PGVs were seen. In both groups, approximately one-third of patients referred to medical genetics did not undergo genetic testing. This study reinforces the importance for patients with CRC to undergo genetic testing, especially those with eoCRC.

**Abstract:**

Over the past 20 years, rates of early-onset colorectal cancer (eoCRC), defined as <50 years of age at diagnosis, have increased, with 16–25% associated with a pathogenic germline variant (PGV) resulting in a hereditary cancer syndrome. In the present study, we sought to further characterize PGVs observed in patients with eoCRC. We conducted a retrospective analysis of patients with a history of CRC referred for genetic counseling at Mayo Clinic Rochester between April 2019 and April 2022. Three hundred and three CRC patients were referred to medical genetics, including 124 with a history of eoCRC. Only 84 patients (68%) with eoCRC referred for genetic counseling completed genetic testing, with an average of 48 genes evaluated. PGVs were identified in 27.4% with eoCRC, including 8.3% with Lynch syndrome (LS). Other detected PGVs known to increase the risk of CRC included *MUTYH* (4.8%), *CHEK2* (3.6%), *APC*, *BMPR1A*, and *TP53* (1.3% each). Among those with aoCRC, 109 patients (61%) completed genetic testing, among which 88% had either a dMMR tumor, personal history of an additional LS malignancy, or family history of LS malignancy, with PGVs detected in 23% of patients. This study reinforces the importance for all patients with CRC, especially those with eoCRC, to undergo germline testing.

## 1. Introduction

Colorectal cancer (CRC) is the third most common malignancy globally and is the second most common cause of cancer-related death [1]. Over the last twenty years, rates of CRC among adults age ≥50 have declined; however, the incidence of early-onset colorectal cancer (eoCRC), defined as CRC diagnosed in those <50 years of age, has been increasing in the United States and other high-income countries and may account for 10% of new colon cancer diagnoses and 25% of new rectal cancer diagnoses in the next 10 years [2,3,4,5]. Early-onset CRC is more likely to be left-sided and later stage at the time of diagnosis, with poorer histopathologic features [4,6,7,8]. The etiology of increased rates of eoCRC is unclear but contributing factors may include changes in the gut microbiome, a Western diet, obesity, antibiotic exposure, and hereditary cancer syndromes [5,6,9,10]. As a result, CRC screening guidelines have adapted and in its 2021 update, the United States Preventative Services Task Force (USPSTF) recommended CRC screening in adults aged 45–49 years, whereas prior recommendations had only suggested initiating screening in those 50 and older in an average risk population [11].

One-third of cases of colorectal cancer have been associated with increased familial risk, with 2–10% attributed to a defined genetic syndrome [12,13,14]. Genetic syndromes associated with increased risk of CRC include Lynch syndrome, as well as polyposis syndromes including familial adenomatous polyposis, *MUTYH*-associated polyposis, Peutz–Jeghers syndrome, or juvenile polyposis syndrome. Among patients with eoCRC, pathogenic germline variants (PGVs) have been reported in as many as 16–25% of cases, with Lynch syndrome accounting for up to half of these [15,16,17,18]. In light of this, the National Comprehensive Cancer Network (NCCN) guidelines recommend multigene panel testing for all patients with eoCRC [19].

At present, we are beginning to understand the spectrum of PGVs observed in eoCRC, with a significant range reported in the frequency and spectrum of germline mutation. In the present study, we evaluate patients with a history of CRC referred to medical genetics at a tertiary medical center and report individual patient data with regard to PGVs.

## 2. Materials and Methods

We conducted an analysis of patients referred to Medical Genetics at Mayo Clinic Rochester between 27 April 2019 and 26 April 2022. Patients were included who had a history of CRC and were further subcategorized based on age at the time of diagnosis, <50 (eoCRC) or ≥50 (average-onset colorectal cancer (aoCRC)). We reviewed medical records and documented patient demographics, personal or family history of malignancy, location and stage of the tumor, tumor molecular characteristics, whether or not the patient underwent germline testing, and if so, the number of genes evaluated and results of this testing.

This study was reviewed and approved (exempt) by the Mayo Clinic Institutional Review Board. 

## 3. Results

### 3.1. Patient Demographics and Tumor Characteristics

Three hundred three patients with a history of CRC were referred to Medical Genetics at Mayo Clinic Rochester between 27 April 2019 and 26 April 2022. This included 124 patients with eoCRC and 179 with aoCRC. Among these, 193 patients elected to pursue germline genetic screening with results available, including 84 with eoCRC and 179 with aoCRC (Figure 1). 

Among those with eoCRC, the median age was 42 years, 59% were male, and 86% were white, non-Hispanic, whereas those with aoCRC had a median age of 62 years, 49% were male, and 93% were white, non-Hispanic (Table 1). Sixty percent and 66% of patients were overweight or obese in the eoCRC and aoCRC groups, respectively. Sixty-nine percent of patients with eoCRC had a left-sided malignancy whereas only 49% of malignancies were left-sided among cases of aoCRC, and 58% had stage III or IV disease at the time of diagnosis in cases of eoCRC, whereas 42% were stage III or IV in cases of aoCRC. 

### 3.2. Tumor Somatic Mutations

Somatic *BRAF* mutations were more frequently seen in the aoCRC group (13% vs. 2%), and *KRAS* mutations were comparable between groups, with 22% and 19% seen in the eoCRC and aoCRC groups, respectively (Table 2). Among *KRAS* mutations, G12D was the most frequently observed alteration (Appendix A). Ten patients (8%) with eoCRC had tumors deficient in mismatch repair (dMMR) by immunohistochemistry (IHC). Fifty patients (28%) with aoCRC had dMMR by IHC, with 46% of these resulting from MLH1/PMS2 loss with BRAF V600E or MLH1 promoter hypermethylation. 

### 3.3. Patients with eoCRC Who Completed Germline Testing

Eighty-six patients with eoCRC who were referred to genetic counseling completed germline testing, with results available for 84 of these patients. Comparing those with eoCRC who underwent genetic testing with results available to those who did not undergo genetic testing, the mean age was comparable at 41.08 and 40.48 years, respectively (Appendix A). Males made up a greater proportion of those who did not complete genetic testing, accounting for 65% not completing testing versus 56% who did complete testing. Comparisons among racial groups are unable to be made due to the low representation of minority groups in this study. 

### 3.4. Pathogenic Germline Variants

Of 84 patients with eoCRC who underwent germline testing with results available, with an average of 47.8 (IQR 30, 56) genes evaluated, 23 (27.4%) had an identifiable PGV, whereas 109 patients with aoCRC underwent germline testing, with an average of 48.9 (IQR 36, 58) genes evaluated, with 25 (23%) having a germline PGV (Table 3). Among those with eoCRC, 17 (20.2%) had a PGV known to increase the risk of CRC while 6 (7%) had a PGV not typically associated with CRC. Among those with aoCRC, 22 (20.2%) had a PGV associated with an increased risk of CRC, while 5 (5%) had a PGV not typically associated with CRC. In addition to PGVs, 2 patients with eoCRC and 4 patients with aoCRC were found to have likely pathogenic germline mutations. Among patients with eoCRC and aoCRC, 8.3% and 9.2% had Lynch syndrome, respectively. Additional PGVs, as well as their associated syndrome/malignancies, are listed in Table 3. In addition to PGVs, 25 (29.8%) and 33 patients (30.3%) with eoCRC and aoCRC, respectively, had an identified variant of uncertain significance (VUS), some of which were detected in patients with an identified PGV (Appendix A). Among patients with eoCRC, 10 had dMMR, among which 9 underwent germline testing, with 3 (33%) of these having a PGV. With aoCRC, 50 patients had dMMR, 34 underwent germline testing, with 9 (25%) having a PGV (including 1 patient with *HOXB13* increased risk allele for prostate cancer). The breakdown of which pathogenic mutation was observed based on MMR protein deficiency, considering BRAF V600E and MLH1 promoter hypermethylation status, is detailed in Table 4.

In the aoCRC group, 96 of 109 patients who underwent germline testing had either a tumor with dMMR, personal history of an additional Lynch syndrome malignancy, or a family history of a Lynch syndrome malignancy. All 25 PGVs identified were found in this cohort with risk factors. Table 5 describes rates of detection of PGVs among patients with various risk factors for a hereditary cancer syndrome. Among patients with eoCRC, 65 had no family history of CRC, of which 40 underwent germline testing and 8 had a PGV (with an additional patient having a likely PGV). Among those with aoCRC, 98 had no family history of colorectal cancer, of which 52 underwent germline testing, and 10 had a PGV detected (with an additional patient having a likely PGV).

## 4. Discussion

In the present study, we report PGVs in 27.4% of patients with eoCRC (29.8% if including likely pathogenic germline mutations), which is comparable to rates previously reported, although on the higher end of this range [15,16,17,18]. Lynch syndrome was found in 8.3% of patients with eoCRC, which is consistent with previously reported rates of approximately 8–13% [15,16,17]. Additional PGVs associated with increased risk of CRC included *MUTYH* (4.8%), *CHEK2* (3.6%), *APC*, *BMPR1A*, and *TP53* (1.3% each).

Detection of these PGVs can have significant implications for the patient and their family members. The most frequently observed PGVs among patients in this study were collectively found in Lynch syndrome genes. A diagnosis of Lynch syndrome conveys an increased risk of other malignancies, such as endometrial and ovarian cancer, and in those diagnosed with CRC, the decision to pursue prophylactic hysterectomy with salpingo-oophorectomy may be made at the time of surgical intervention for CRC [19]. Additionally, family members would require testing for Lynch syndrome, as the presence of a PGV would require referral to genetic counseling to discuss specific recommendations for earlier cancer screening, such as beginning colonoscopies at age 20–25 in those with MLH1/MSH2 PGVs [19]. Other PGVs known to increase the risk of CRC included biallelic *MUTYH,* which was seen in 2.4% and 1.8% of those with eoCRC and aoCRC collectively and is associated with *MUTYH*-associated polyposis, which confers a 70% risk of developing CRC by age 70 [20]. NCCN guidelines recommend germline testing for family members at risk, and if found to have biallelic *MUTYH* mutation, initiating earlier CRC screening and depending on polyp burden, considering prophylactic colectomy [19]. As such, germline screening for genes related to CRC has implications for the management of the patient, but also for family members.

In addition to genes increasing the risk of CRC, we also report PGVs not typically associated with increased risk of CRC in 7.1% of eoCRC patients, including *ATM*, *BRCA1*, *PALB2*, *SDHA*, and *MITF*. A similar phenomenon was reported by Pearlman et al., who found that 18.2% of eoCRC patients had a PGV not typically associated with CRC, including *ATM*, *CHEK2* (classified as a less likely cause of CRC in this study), *BRCA1*, *CDKN2A*, and *PALB2* [15]. The significance of this remains unclear at this time, and as previously noted, with increased frequency of large panel germline testing, more associations between PGVs and CRC may be recognized. Additionally, while the detection of these PGVs may not have implications with regard to their colon cancer diagnosis, it may impact future cancer screening for patients and their family members. For example, *BRCA1*, found in one patient with eoCRC, confers an increased risk of breast, ovarian, prostate, and pancreatic cancer. As such, a patient with a *BRCA1* mutation undergoing surgical management of colorectal cancer may opt to pursue prophylactic hysterectomy with salpingo-oophorectomy at the same time considering this mutation. With regards to additional cancer screening for the patient, and other family members found to harbor this PGV, NCCN guidelines recommend more intensive screening, including initiation of annual breast MRI beginning at age 25 [21]. Therefore, even seemingly unrelated PGVs may have a significant impact on the patient and their family members.

When comparing rates of PGVs in eoCRC to those observed in aoCRC, we see comparable frequency, with similar distribution of impacted genes. However, direct comparison is challenging as the aoCRC cohort in this study is a highly selected group who was referred to medical genetics for a specific reason, whether it be due to the dMMR status of their tumor, personal history of an additional malignancy (56% had a personal history of an additional malignancy, whereas only 26% with eoCRC had a personal history of an additional malignancy), or a strong family history of malignancy. When comparing to other studies looking at rates of PGVs in patients with CRC irrespective of age at diagnosis, rates have been lower, ranging from 10–15%, which further demonstrates the selected nature of the aoCRC population in this study [14,22].

Current NCCN guidelines recommend multigene panel testing in patients with CRC who are diagnosed at age <50, have a tumor with dMMR, have a history of synchronous or metachronous Lynch syndrome malignancies, have a first- or second-degree relative with a Lynch syndrome malignancy diagnosed at age <50, have two or more first- or second-degree relatives with a Lynch syndrome malignancy diagnosed at any age, or have a PREMM5 score greater than 2.5% [19]. These guidelines also note that multigene panel testing may be considered in all patients with CRC diagnosed at age ≥50. However, there has been much debate regarding the utility of universal germline testing for all patients with CRC. Pearlman et al. evaluated screening only CRC patients with dMMR tumors, eoCRC, synchronous or metachronous Lynch syndrome malignancies, or with a first-degree relative with a Lynch syndrome malignancy, and found that screening by these criteria failed to detect 38.6% of PGVs [23]. Uson et al. found that 9.4% of patients with CRC had a PGV that would not have been detected by guideline-based screening or by CRC-specific gene panels [22].

However, whether or not the additional PGVs detected by screening all patients beyond these guidelines may impact patient management or improve survival remains to be determined. In our study, only 69% of patients with eoCRC referred to medical genetics ultimately underwent some form of germline testing, while 61% with aoCRC completed testing. Of those with aoCRC who underwent genetic testing, 88% had a dMMR tumor, personal history of an additional Lynch syndrome malignancy, or some family history of a prior Lynch syndrome malignancy. This group with high-risk features captured all PGVs among individuals with aoCRC included in this study. At this time, further study is needed to evaluate the utility of expanded germline testing, and the implications for patient outcomes.

When looking more closely at patients with dMMR by IHC, loss of MLH1/PMS2 in the setting of a BRAF V600E mutation or MLH1 promoter hypermethylation, suggestive of a sporadic mutation, was more frequently seen in patients with aoCRC. Among all patients with CRC who had a loss of MLH1/PMS2 with BRAF V600E or MLH1 promoter hypermethylation, 2 of 16 were ultimately found to have a pathogenic germline mutation, but neither of these patients had Lynch syndrome, with one having an increased risk allele for prostate cancer. It is of interest that among patients with loss of other MMR proteins, 66% were found to have Lynch syndrome, which demonstrates the importance of germline analysis in this group.

A limitation of this study is selection bias as patients with greater risk factors for a hereditary cancer syndrome may be more likely to pursue germline testing. This is certainly true for patients with aoCRC, and to a smaller extent, patients with eoCRC. In spite of this, this institutional study contributes to our current understanding of rates of PGVs in early-onset colorectal cancer, as well as the spectrum of mutations that are observed.

## 5. Conclusions

In recent history, rates of eoCRC have been increasing and will make up a significant portion of new CRC diagnoses in the next ten years. Current NCCN guidelines recommend germline testing in all patients with eoCRC, and PGVs may be detected in as many as 25% of these patients. However, despite these recommendations, in the current study, nearly 1/3 of patients with eoCRC referred to genetic counseling ultimately did not pursue genetic testing. As demonstrated, the PGVs detected among this cohort can have significant implications for management for the patient, as well as their family members, and as such, demonstrates a need to increase adherence to screening guidelines. This study demonstrates the importance for patients with CRC to undergo genetic testing, especially those with eoCRC, as well as the need for further research evaluating reasons why individuals with CRC decline to pursue genetic testing.

## Figures and Tables

**Figure 1 cancers-15-03570-f001:**
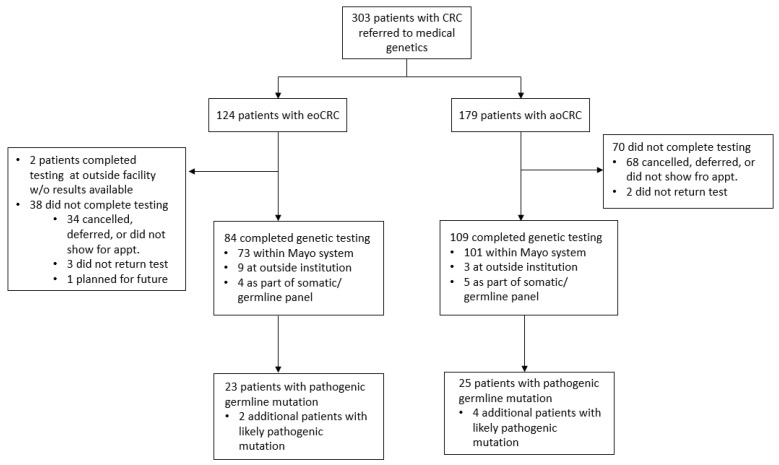
Consort diagram of patients in this study. Abbreviations: CRC: colorectal cancer; eoCRC: early-onset colorectal cancer; aoCRC: average-onset colorectal cancer.

**Table 1 cancers-15-03570-t001:** Patient demographics by age of first CRC diagnosis.

	All (*n* = 303)	eoCRC (*n* = 124)	aoCRC (*n* = 179)
Age, median (IQR)	51 (44, 65)	42 (38, 46)	62 (52, 70.5)
Sex			
Male	160 (53)	73 (59)	87 (49)
Female	143 (47)	51 (41)	92 (51)
Race			
White, non-Hispanic	274 (90)	107 (86)	167 (93)
Black	5 (2)	3 (2)	1 (1)
Hispanic	7 (2)	2 (2)	5 (3)
Asian	6 (2)	3 (2)	3 (2)
Am. Indian/Alaska Native	2 (1)	2 (2)	0 (0)
Other/Unknown	10 (3)	7 (6)	3 (2)
BMI near diagnosis			
<25	76 (25)	33 (27)	43 (24)
≥25, <30	74 (24)	31 (25)	43 (24)
≥30	120 (40)	44 (35)	76 (42)
Unknown	33 (11)	16 (13)	17 (9)
Tumor location			
Right	118 (39)	32 (26)	86 (48)
Left	174 (57)	86 (69)	88 (49)
Rectum/rectosigmoid	70 (23)	36 (29)	34 (19)
Unknown	11 (4)	6 (5)	5 (3)
Stage at diagnosis			
0	9 (3)	6 (5)	3 (2)
1	63 (21)	13 (10)	50 (28)
2	66 (22)	25 (20)	41 (23)
3	95 (31)	44 (35)	51 (28)
4	53 (17)	28 (23)	25 (14)
Unknown	17 (6)	8 (6)	9 (5)
Personal history of malignancy *			
All	102 (34)	28 (23)	74 (41)
Synchronous (colon)	6 (2)	1 (1)	5 (3)
Metachronous (colon)	12 (4)	6 (5)	6 (3)
Breast	20 (7)	6 (5)	14 (8)
Ovary	3 (1)	0 (0)	3 (2)
Uterus	6 (2)	0 (0)	6 (3)
No/unknown	201 (66)	96 (77)	105 (59)
Family history of malignancy **			
All	274 (90)	112 (90)	162 (91)
Colon	140 (46)	59 (48)	81 (45)
Breast	135 (45)	49 (40)	86 (48)
Ovary	22 (7)	7 (6)	15 (8)
Uterus	15 (5)	9 (7)	6 (3)
Pancreas	28 (9)	12 (10)	16 (9)
No/unknown	29 (10)	12 (10)	17 (9)

Abbreviations: eoCRC: early-onset colorectal cancer; aoCRC: average-onset colorectal cancer; BMI: body mass index. * Among patients with aoCRC, 2 patients had a history of prior colon cancer and breast cancer, 1 had a history of breast and ovarian cancer, 2 had a history of breast and uterine cancer, and 1 had ovarian and uterine cancer. ** Among patients with eoCRC, 38 had a family history of 2 or more different Lynch syndrome malignancies, and among patients with aoCRC, 59 had a family history of 2 or more different Lynch syndrome malignancies.

**Table 2 cancers-15-03570-t002:** Somatic mutations by age of first CRC diagnosis.

	eoCRC (*n* = 124)	aoCRC (*n* = 179)
BRAF		
WT	70 (56)	89 (50)
V600E	3 (2)	24 (13)
G469R	0 (0)	1 (1)
Unknown	51 (41)	65 (36)
*KRAS*		
WT	45 (36)	68 (38)
Mutant	27 (22)	34 (19)
Unknown	52 (42)	77 (43)
NRAS		
WT	66 (53)	97 (54)
Mutant	2 (2)	3 (2)
Unknown	56 (45)	79 (44)
Mismatch repair		
pMMR	91 (73)	109 (61)
dMMR	10 (8)	50 (28)
MLH1/PMS2 loss, BRAF V600E or MLH1 promoter hypermethylation	1 (1)	23 (13)
MLH1/PMS2 loss, BRAF WT and MLH1 hypermethylation negative	4 (3)	5 (3)
MLH1/PMS2 loss, without BRAF or MLH1 hypermethylation test result	2 (2)	12 (7)
Other dMMR	3 (2)	9 (5)
Unknown dMMR	0 (0)	1 (1)
**Unknown**	23 (19)	20 (11)

Abbreviations: eoCRC: early-onset colorectal cancer; aoCRC: average-onset colorectal cancer; WT: wild-type; pMMR: proficient mismatch repair; dMMR: deficient mismatch repair.

**Table 3 cancers-15-03570-t003:** Pathogenic and likely pathogenic variants observed by age of first CRC diagnosis.

	Syndrome or Cancer(s) Associated with Gene	eoCRC (*n* = 84)	aoCRC (*n* = 109)
Any		23 (27.4)	25 (23)
*MLH1*	Lynch syndrome	3 (3.6)	2 (1.8)
*MSH2*		2 (2.4)	3 (2.8)
*PMS2*		1 (1.2)	3 (2.8)
*MSH6*		1 (1.2)	2 (1.8)
*MUTYH* (biallelic)	Colorectal	2 (2.4)	2 (1.8)
*MUTYH* (monoallelic) *		2 (2.4)	2 (1.8)
*PTEN*	Colorectal, kidney, thyroid, melanoma, breast, uterus	0 (0)	2 (1.8)
*APC*	Familial adenomatous polyposis	1 (1.2)	0 (0)
*BMPR1A*	Juvenile polyposis syndrome	1 (1.2)	0 (0)
*NTHL1* (monoallelic) **	Colorectal, breast	0 (0)	1 (0.9)
*TP53*	Li–Fraumeni syndrome	1 (1.2)	0 (0)
*CHEK2*	Breast, colorectal	3 (3.6)	5 (4.6)
*ATM*	Breast, pancreas, prostate, ovary	3 (3.6)	2 (1.8)
*BRCA1*	Breast, uterus, ovary, prostate, pancreas	1 (1.2)	0 (0)
*PALB2*	Breast, ovary, pancreas	1 (1.2)	0 (0)
*SDHA*	GIST, paraganglioma, pheochromocytoma, renal cell carcinoma	1 (1.2)	1 (0.9)
*MITF*	Melanoma, kidney	0 (0)	1 (0.9)
*HOXB13 ****	Prostate	0 (0)	1 (0.9)
Likely Pathogenic			
Any		2 (2.4)	4 (3.7)
*MSH2*	Lynch syndrome	0 (0)	2 (1.8)
*PMS2*		0 (0)	1 (0.9)
*FH*	Kidney	1 (1.2)	0 (0)
*CDH1*	Stomach, breast	1 (1.2)	0 (0)
*NBN* (monoallelic)	Nijmegen breakage syndrome	0 (0)	1 (0.9)

Abbreviations: eoCRC: early-onset colorectal cancer; aoCRC: average-onset colorectal cancer; GIST: gastrointestinal stromal tumor. Notes: 2 patients with aoCRC had 2 different pathogenic variants; 1 patient with eoCRC and 2 patients with aoCRC had both a pathogenic and likely pathogenic variant. * Possible increased risk of colorectal cancer. ** Questionable associations in monoallelic state. *** Increased risk allele.

**Table 4 cancers-15-03570-t004:** Pathogenic variant detection in patients with eoCRC and aoCRC based on dMMR status.

	dMMR Status	*n*	with PGV	PGVs Observed
eoCRC	MLH1/PMS2 loss, BRAF V600E or MLH1 promoter hypermethylation	1	1	*CHEK2*
	MLH1/PMS2 loss, BRAF WT and MLH1 hypermethylation negative	4	1	*MUTYH* (biallelic)
	MLH1/PMS2 loss, without BRAF or MLH1 hypermethylation test result	2	0	N/A
	Other dMMR	2	1	*MSH2*
aoCRC	MLH1/PMS2 loss, BRAF V600E or MLH1 promoter hypermethylation	15	1	*HOXB13* *
	MLH1/PMS2 loss, BRAF WT and MLH1 hypermethylation negative	3	1	*MLH1*
	MLH1/PMS2 loss, without BRAF or MLH1 hypermethylation test result	8	2	*MSH6*, *ATM*
	Other dMMR	7	5	*PMS2* (2), *MSH2* (2), *MSH6*
	Unknown dMMR	1	0	N/A

Abbreviations: eoCRC: early-onset colorectal cancer; aoCRC: average-onset colorectal cancer; dMMR: deficient mismatch repair; PGV: pathogenic variant. * Increased risk allele for prostate cancer.

**Table 5 cancers-15-03570-t005:** Percentage of patients found to have a pathogenic germline mutation by varying risk factors.

	eoCRC	aoCRC
dMMR tumor	3/9 (33)	8/34 (24)
Personal history of additional LS malignancy	4/9 (44)	8/21 (38)
Family history of LS malignancy	19/61 (31)	19/85 (22)
One of the above	20/64 (31)	25/96 (26)

Abbreviations: dMMR: deficient mismatch repair; LS: Lynch syndrome; eoCRC: early-onset colorectal cancer; aoCRC: average-onset colorectal cancer.

## Data Availability

Data is not available.

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
