# Peer review of "Completion of Genetic Testing and Incidence of Pathogenic Germline Mutation among Patients with Early-Onset Colorectal Cancer: A Single Institution Analysis"

_cancers, 2023, doi:10.3390/cancers15143570_

Round 1

Reviewer 1 Report

This study provides a comprehensive overview of the frequency and results of germline genetic testing of patients with early- and average-onset colorectal cancer, based on real-life data. I have very little to add or suggest, as the results mostly speak for themselves. I do however recommend to add some paragraphs on the implications on prognosis and treatment of specific germline mutations, as far as these are known. Only by taking those into account, the true utility of the testing efforts can be appropriately judged.

Author Response

We greatly appreciate the reviewer’s comments and we have added specific examples of germline mutations, both related and unrelated to colorectal cancer, and how they may impact management of the patient and their family members, giving further support to the claim that universal germline screening has benefit and should be considered in all patients with CRC.

Reviewer 2 Report

First of all, I would like to congratulate you on your extensive work on the study of mutations in colon cancer. It is a large number of patients. However, the conclusions of the study are obvious. 

I think it is necessary to try to give a new message to the conclusions. What is new in the work that has been done? What is the most important message of the study?

The cases of Lynch syndrome should be separated, as it is clear that it is an entity with already defined genetic markers.

It would be of interest to separate those cases that are sporadic, without any family history. 

What are the clinical repercussions of the presence of germline genetic mutations and their association with somatic mutations? 

Author Response

Comment 1:

I think it is necessary to try to give a new message to the conclusions. What is new in the work that has been done? What is the most important message of the study?

Response 1:

Thank you for the reviewer’s suggestion. We have added a sentence to the conclusion noting how the PGVs detected among this population had potential to impact both cancer management and future cancer screening for the patient and their family members. We further emphasize the point that despite this potential benefit, only 2/3 of those with eoCRC pursued genetic screening, which we feel is a novel point to note. We have added to our conclusion that considering this point, future study is needed to evaluate why patients with eoCRC choose not to pursue germline testing.

Comment 2:

The cases of Lynch syndrome should be separated, as it is clear that it is an entity with already defined genetic markers.

Response 2:

We completely agree with the reviewer that Lynch syndrome and the role of PGVs associated with Lynch syndrome are well defined. The prognostic and predictive values of Lynch syndrome have been seen in large studies.

Comment 3:

It would be of interest to separate those cases that are sporadic, without any family history. 

Response 3:

We appreciate the reviewer’s recommendation and we did further work as suggested. Among those without any family history of malignancy, for those with aoCRC, only 5 patients underwent testing and among those with eoCRC, only 4 underwent genetic testing. Therefore, we have not added this analysis as this population was quite small.

For those with no known family history of CRC, among those with eoCRC, 65 had no family history of CRC, 40 underwent germline testing, and 8 had a PGV with one having a likely PGV. Among those with aoCRC, 98 had no family history of colorectal cancer, 52 underwent germline testing, and 10 had a PGV and 1 had a likely PGV. This had been added to the results section of the manuscript.

Comment 4:

What are the clinical repercussions of the presence of germline genetic mutations and their association with somatic mutations? 

Response 4:

We have added more information regarding the clinical repercussions of specific PGVs, giving strength to the argument that all should undergo germline screening. This has been added to the discussion section and was also requested by another reviewer. With regards to the association with somatic mutations, this was out of the scope of this study and we are unable to provide this analysis at this time.

Round 2

Reviewer 2 Report

I agree with the comments and changes made by the author.